# FREQUENCY REGULARIZED DEEP CONVOLUTIONAL DICTIONARY LEARNING AND APPLICATION TO BLIND DENOISING

## ABSTRACT

Sparse representation via a learned dictionary is a powerful prior for natural images. In recent years, unrolled sparse coding algorithms (e.g. LISTA) have proven to be useful for constructing interpretable deep-learning networks that perform on par with state-of-the-art models on image-restoration tasks. In this study we are concerned with extending the work of such convolutional dictionary learning (CDL) models. We propose to construct strided convolutional dictionaries with a single analytic low-pass filter and a set of learned filters regularized to occupy the complementary frequency space. By doing so, we address the necessary modeling assumptions of natural images with respect to convolutional sparse coding and reduce the mutual coherence and redundancy of the learned filters. We show improved denoising performance at reduced computational complexity when compared to other CDL methods, and competitive results when compared to popular deep-learning models. We further propose to parameterize the thresholds in the soft-thresholding operator of LISTA to be proportional to the estimated noise-variance from an input image. We demonstrate that this parameterization enhances robustness to noise-level mismatch between training and inference.

## 1 INTRODUCTION

Sparsity in a transform domain is an important and widely applicable property of natural images. This property can be exploited in a variety of tasks such as signal representation, feature extraction, and image processing. For instance, consider restoring an image from a degraded version (noisy, blurry, or missing pixels). These inverse problems are generally ill-posed and require utilizing adequate prior knowledge, for which sparsity has proven extremely effective (Mairal et al., 2014).

In recent years, such problems have been tackled with deep neural network architectures that achieve superior performance but are not well-understood in terms of their building blocks. In this study, we are interested in utilizing the knowledge from classical signal processing and spare coding literature to introduce a learned framework which is interpretable and that can perform on-par with state-of-the-art deep-learning methods. We choose to explore this method under the task of natural image denoising, in line with much of the recent literature (Sreter & Giryes, 2018; Simon & Elad, 2019; Lecouat et al., 2020). As a benefit of this interpretability, we are able to extend the framework for a blind-denoising setting using ideas from signal processing.

In sparse representation we seek to approximate a signal as a linear combination of a few vectors from a set of vectors (usually called dictionary atoms). Olshausen & Field (1996), following a neuroscientific perspective, proposed to adapt the dictionary to a set of training data. Later, dictionary learning combined with sparse coding was investigated in numerous applications (Mairal et al., 2009a; Protter & Elad, 2008). More specifically, for a set of $N$ image patches (reshaped into column vectors) $\boldsymbol{X} = [\boldsymbol{x}_1, \cdots, \boldsymbol{x}_N] \in \mathbb{R}^{m \times N}$, we seek to find the dictionary $\boldsymbol{D}^* \in \mathbb{R}^{m \times k}$ and the sparse representation $\boldsymbol{Z}^* = [\boldsymbol{z}_1^*, \cdots, \boldsymbol{z}_N^*] \in \mathbb{R}^{k \times N}$ such that

$$\boldsymbol{D}^*, \boldsymbol{Z}^* = \arg\min_{\boldsymbol{D}, \boldsymbol{Z}} \sum_{i=1}^{N} \|\boldsymbol{z}_i\|_0 \text{ subject to: } \boldsymbol{D}\boldsymbol{z}_i = \boldsymbol{x}_i, \ \forall i = 1, \cdots, N. \tag{1}$$

This formulation is not tractable for large signals since minimizing the $\ell_0$-pseudo-norm involves a combinatorial optimization (Natarajan, 1995). To address this complication, a popular technique is to relax the problem by using the $\ell_1$-norm as a surrogate (Sreter & Giryes, 2018). When dealing with inverse problems such as denoising, learning the dictionary from the degraded signal has shown effective (Toić & Frossard, 2011). Let $\boldsymbol{y}_i = \boldsymbol{x}_i + \boldsymbol{n}_i \in \mathbb{R}^m$ represent the noisy signal where $\boldsymbol{n}_i$ follows an additive white Gaussian distribution, $\mathcal{N}\left(\boldsymbol{0}, \sigma_n^2 \boldsymbol{I}\right)$. Then, the relaxed formulation can be written as

$$\min_{\boldsymbol{D},\boldsymbol{Z}} \sum_{i=1}^{N} \|\boldsymbol{z}_i\|_1 \text{ s.t. } \sum_{i=1}^{N} \frac{1}{2} \|\boldsymbol{D}\boldsymbol{z}_i - \boldsymbol{y}_i\|_2^2 \leq \epsilon \text{ or } \min_{\boldsymbol{D},\boldsymbol{Z}} \sum_{i=1}^{N} \frac{1}{2} \|\boldsymbol{D}\boldsymbol{z}_i - \boldsymbol{y}_i\|_2^2 + \lambda \|\boldsymbol{z}_i\|_1 \qquad (2)$$

where $\lambda$ is a regularization parameter and is nontrivialy related to the representation error $\epsilon$. We will refer to this as the basis-pursuit denoising (BPDN) formulation of dictionary learning. Many iterative algorithms have been proposed in the literature to solve this problem (Mairal et al., 2014). A majority of these algorithms split the problem into a step updating the dictionary followed by a step solving for the sparse codes.

Note that learning a dictionary over independent image patches neglects the dependencies between these patches. As a result, the models involving patch processing are inherently sub-optimal (Batenkov et al., 2017; Simon & Elad, 2019). Although enforcing local priors on merged images (Sulam & Elad, 2015) and utilizing self-similarity between patches (Mairal et al., 2009b) have been proposed as ideas to mitigate this flaw, ideally a global shift-invariant model is more appropriate. By constraining the dictionary to have a Toeplitz structure, the Convolutional Sparse Coding (CSC) model has been introduced which replaces the local patch processing with a global convolution (Grosse et al., 2007; Papyan et al., 2017).

Algorithms for solving the CSC model are also discussed in (Moreau & Gramfort, 2019; Wohlberg, 2017). In this study, we are interested in interpretable CSC-based deep-learning models. A metric known as the mutual-coherence is well known to be related to the representation capability of the dictionary and is of special concern in using the CSC model with natural images (Simon & Elad, 2019). We take an alternative route to Simon & Elad (2019) in addressing the mutual-coherence of CSC-based deep-learning models, which is both less computationally expensive and improves the denoising performance. We continue the discussion about CSC-based deep-learning models in Sec. 1.1.

Another important aspect of the sparse representation is the sparse coding algorithm. For a given signal $\boldsymbol{y} \in \mathbb{R}^m$ and dictionary $\boldsymbol{D}$, iterative soft-thresholding algorithm (ISTA) (Beck & Teboulle, 2009) finds the solution to the BPDN functional, $\boldsymbol{z}^* = \arg\min_{\boldsymbol{z}} 1/2 \|\boldsymbol{D}\boldsymbol{z} - \boldsymbol{y}\|_2^2 + \lambda \|\boldsymbol{z}\|_1$, by repeating the following iteration until a convergence criterion is reached:

$$\boldsymbol{z}^{(k+1)} = S_{\lambda\eta^{(k)}} \left( \boldsymbol{z}^k - \eta^{(k)} \boldsymbol{D}^T \left( \boldsymbol{D}\boldsymbol{z}^{(k)} - \boldsymbol{y} \right) \right) \text{ where } S_\theta(x) = \text{sgn}(x)(|x| - \theta)_+, \quad \theta \geq 0. \quad (3)$$

Here, $\eta^{(k)}$ is the step-size of the descent algorithm at iteration $k$. Note that performing sparse coding with an iterative method like ISTA for all patches is computationally exhausting and slow. To resolve this issue, Gregor & LeCun (2010) proposed to approximate the sparse coding via a learned differentiable encoder, dubbed LISTA. Further extensions of LISTA both in terms of practice and theory have been studied in the literature (Wu et al., 2019; Chen et al., 2018). More recently, using LISTA combined with dictionary learning has been a research highlight (Sreter & Giryes, 2018; Simon & Elad, 2019; Lecouat et al., 2020). We refer to this type of models that leverages LISTA for convolutional dictionary learning as CDL models.

## 1.1 RELATED WORKS

In this study, we are interested in the CDL model that concatenates a LISTA network with a linear convolutional synthesis dictionary. Let $\boldsymbol{D}$ be a convolutional dictionary with $M$ filters (and their integer shifts). We denote the filters in $\boldsymbol{D}$ by $\boldsymbol{d}^j$ where $j \in \{1, \cdots, M\}$. Let $\boldsymbol{Z}_i$ denote the sparse code for the data sample $\boldsymbol{y}_i = \boldsymbol{x}_i + \boldsymbol{n}_i$ where $i \in \{1, 2, \cdots, N\}$ and $\boldsymbol{n} \sim \mathcal{N}(0, \sigma_n^2 \boldsymbol{I})$. The corresponding subband signal to $\boldsymbol{d}^j$ in $\boldsymbol{Z}_i$ can be denoted as $\boldsymbol{z}_i^j$. Then the convolutional dictionary learning problem is written as

$$\underset{\boldsymbol{d}^j, \boldsymbol{Z}_i}{\text{minimize}} \sum_{i=1}^{N} \frac{1}{2} \|\boldsymbol{y}_i - \sum_{j=1}^{M} \boldsymbol{d}^j * \boldsymbol{z}_i^j\|_2^2 + \lambda \sum_{j=1}^{M} \|\boldsymbol{z}_i^j\|_1. \qquad (4)$$

Sreter & Giryes (2018) introduce the approximate convolutional sparse coding (ACSC) framework for "task-driven convolutional sparse coding", combining a convolutional extension of LISTA with a linear convolutional decoder. The proposed framework offers a strategy for training an approximate convolutional sparse coding network and a corresponding convolutional dictionary in an end-to-end fashion. They demonstrate competitive performance against classical patch-based methods such as K-SVD (Aharon et al., 2006), on image denoising and image inpainting. Our proposed baseline model (CDLNet) differs from the ACSC model by use of mean-subtraction preprocessing, employing small-strided convolutions, and imposing a norm-constraint on the synthesis dictionary.

Simon & Elad (2019) extend the framework of Sreter & Giryes (2018) by considering the role of stride in the stable recovery of signals and proposed the "CSCNet" framework. They argue that the CSC model for image representation in a sparse domain is limited by the inclusion of "smooth filters", which are required to represent the piecewise smooth characteristics of natural images. This limitation manifests itself in the maximum cross-correlation between atoms of the dictionary, known as the mutual-coherence. They empirically show that using relatively large stride, while processing shifted-duplicates of the input, improves denoising performance of the model. Although using large stride reduces the mutual coherence of the learned filters, all possible shifts of the image need to be processed and averaged, yielding a model very similar to patch-processing. We propose a frequency regularization strategy to mitigate the problem of smooth-varying filters which does not require shift-averaging.

Note that the parameter $\lambda$ in equation 4 depends on the desired sparsity, relative to the noise-level, and is directly related to the threshold values in ISTA. Sreter & Giryes (2018) propose to learn different thresholds for each channel, effectively changing the regularizer term in equation 4 to $\sum_{j=1}^{M} \|\lambda^j z_i^j\|_1$. Inspired by the benefit of minimax-concave (MC) penalty (Selesnick, 2017) over $\ell_1$ norm, Pokala et al. (2020) propose "ConFirmNet" where firm-thresholding function is used in the network. Kim & Park (2020) propose a signal adaptive threshold scheme for LISTA where the threshold is decreased if the previous estimate of an element is large.

Mohan et al. (2020) explore the role of bias-vectors in popular deep-learning network's convolution operators. They advocate for eliminating the biases completely to improve generalization in blind-denoising where there is mismatch between training and inference noise level. Isogawa et al. (2017) propose altering the biases of deep neural-networks by scaling them with the input noise standard-deviation. Their method is ultimately a non-blind denoising scheme as they use the ground-truth noise statistics during training and inference. In contrast, we propose a blind-denoising scheme that is motivated by the interpretation of the biases in LISTA as thresholds and employ a scaling by the noise variance (in the last layer of LISTA), estimated from the input signal during training and inference. Performance of different denoising techniques on other noise distributions have also been studied in the literature, which is not the focus of this study (Abdelhamed et al., 2018; Plotz & Roth, 2017).

## 1.2 CONTRIBUTION OF THIS STUDY

The unrolled convolutional sparse coding and dictionary learning frameworks have led to the field dubbed "interpretable deep-learning". The networks constructed in such a way have the benefit of interpretability and decreased parameter count while performing quite closely to other state-of-the-art deep-learning models. In this study we further extend such frameworks. We propose utilizing a strided convolutional dictionary with a fixed low-pass channel and a set of frequency-regularized learnt filters (Section 2.2). Our experimental results demonstrate that such frequency regularization and small stride leads to more interpretable dictionary filters than the prior work. Consequently, by limiting the number of low-pass atoms in the dictionary and using small-strided convolutions, we address the modeling assumptions associated with the convolutional sparse coding model (Section 2.1.1). Additionally, leveraging interpretability of our network, we propose to parameterize the soft-thresholding operator in LISTA such that the thresholds are proportional to the estimated input noise-level for a given image (Section 2.3). Experimentally, we show improved denoising performance at reduced computational complexity compared to other frameworks (Section 3.2). Furthermore, our parameterization of the learned thresholds greatly improves robustness to noise-level mismatch between training and inference and increases the generalizability of the network (Section 3.3).

## 2 PROPOSED FRAMEWORK

### 2.1 CONVOLUTIONAL DICTIONARY LEARNING NETWORK (CDLNET)

We seek to solve the natural image denoising problem via the convolutional dictionary learning model on the BPDN functional,

$$\underset{\boldsymbol{d}^j, \boldsymbol{Z}_i}{\text{minimize}} \sum_{i=1}^{N} \frac{1}{2} \|\boldsymbol{y}_i - \sum_{j=1}^{M} \boldsymbol{d}^j * \boldsymbol{z}_i^j\|_2^2 + \sum_{j=1}^{M} \|\lambda^j \boldsymbol{z}_i^j\|_1 \text{ subject to: } \|\boldsymbol{d}^j\|_2^2 \le 1 \,\forall j \in \{1, \cdots, M\}. \quad (5)$$

A norm constraint is imposed on the dictionary atoms to remove the arbitrary scaling of coefficients, as in Mairal et al. (2014). We propose the following learned CDL model, dubbed CDLNet, which involves a LISTA module followed by a learned convolutional synthesis dictionary, $\boldsymbol{D}$,

$$\hat{\boldsymbol{x}} = \boldsymbol{D}\boldsymbol{z}^{(K)}, \quad \boldsymbol{z}^{(k+1)} = S_{\boldsymbol{\theta}^{(k)}}\left(\boldsymbol{z}^{(k)} - \boldsymbol{A}^{(k)}(\boldsymbol{B}^{(k)}\boldsymbol{z}^{(k)} - \boldsymbol{y})\right), \quad k = 0, \dots, K-1, \quad \boldsymbol{z}^{(0)} = \boldsymbol{0} \quad (6)$$

where ISTA has been unrolled for $K$ steps. Here, $\boldsymbol{A}^{(k)}$ and $\boldsymbol{B}^{(k)}$ are small-strided convolution analysis and synthesis operators respectively. We untie the parameters at each iteration of LISTA following the theoretical analysis of Chen et al. (2018). A threshold vector $0 \le \boldsymbol{\theta}^{(k)} \in \mathbb{R}^M$ is learned corresponding to the $M$ subbands of the convolutional sparse code at iteration $k$.

The reconstructed signal is given by $\hat{\boldsymbol{x}}$. The total learnable parameters are given by $\Theta = \{\{\boldsymbol{A}^{(k)}, \boldsymbol{B}^{(k)}, \boldsymbol{\theta}^{(k)}\}_{k=0}^{K-1}, \{\boldsymbol{d}^j\}_{j=1}^{M}\}$. Note that a traditional LISTA network requires supervised training on sparse codes computed from ISTA. On the other hand, the CDLNet can learn to approximate sparse coding and the dictionary in an unsupervised fashion by minimizing a suitable loss function designed for the image reconstruction task (Sreter & Giryes, 2018) (i.e. unsupervised in the code-domain, but supervised in the signal-domain). In this sense the network mimics the common dictionary learning strategy of alternating between computing sparse codes and updating the dictionary, however, the sparse coding is done via a learned algorithm with fast inference. An alternative unsupervised LISTA training strategy, which minimizes the BPDN functional (equation 2), was presented in Ablin et al. (2019). As in (Sreter & Giryes, 2018; Simon & Elad, 2019), we employ $\ell_2$ loss between the restored image and its ground-truth clean image throughout this study.

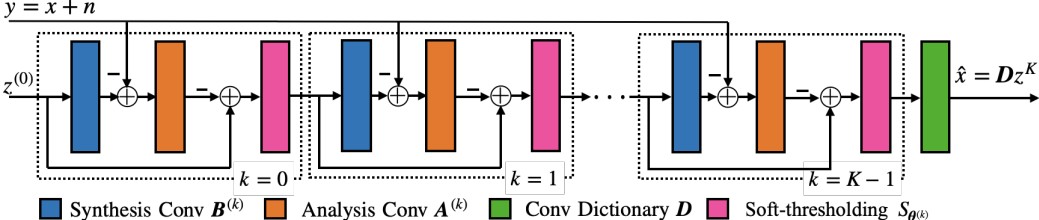

Figure 1: Block diagram of CDLNet.

### 2.1.1 A DISCUSSION ON MUTUAL COHERENCE OF THE LEARNED DICTIONARY

The approximately piecewise smooth nature of natural images will require a synthesis dictionary to contain "smoothly-varying" low-pass atoms. As Simon & Elad (2019) discuss, such low-pass atoms pose a problem for BPDN. A sufficient condition for the faithful recovery of the $\ell_0$ sparse code from an $\ell_1$ basis pursuit can be given in terms of the dictionary's mutual coherence, $\mu(\boldsymbol{D})$. Note that for matrix $\boldsymbol{A}$ with normalized columns $\mathbf{a}_i$, we have $\mu(\boldsymbol{A}) = \max_{i \ne j} |\mathbf{a}_i^\top \mathbf{a}_j|$. For the convolutional dictionary, the atoms of $\boldsymbol{D}$ are composed of the shifts of its filters, $\{\boldsymbol{d}^j\}_{j=1}^{M}$. This poses a problem in that the inner product between any of such low-pass filters and their own integer-translates will greatly increase the mutual coherence and potentially harm the reconstruction performance of the system.

Sreter & Giryes (2018) do not address this issue in the ACSC framework. Simon & Elad (2019) propose to use large strides on the order of the filter size, along with averaging reconstructions from shifted input signals – effectively returning to a patch-based approach. In CDLNet we use small

strided convolutions (stride=2, in both horizontal and vertical directions) without an averaging reconstruction scheme. Furthermore, we use a preset low-pass filter, and parameterize other filters to be in the complimentary frequency space of the low-pass. We empirically show that the combination of the proposed regularization scheme and small stride reduces the mutual coherence of the dictionary, improves denoising performance of the model, and reduces the computational cost.

## 2.2 FREQUENCY REGULARIZATION OF A CONVOLUTIONAL DICTIONARY

In this section we propose a method for regularizing the synthesis dictionary to contain only a single low-pass filter. Note that in the BPDN formulation, the hyperparameter $\lambda$ determines a trade-off between data-fidelity to the observation, $\boldsymbol{y}$, and sparsity of the transform domain coefficients, $\boldsymbol{z}$. Following Sreter & Giryes (2018), we extend this to a vector, $\boldsymbol{\lambda} \in \mathbb{R}^M$, to reflect prior knowledge on the expected levels of sparsity in different subbands of the decomposition. The learned thresholds, $\boldsymbol{\theta}^{(k)}$ ultimately reflect these weights, representing sparsity priors on each subband. In the case of natural image reconstruction, their piecewise smooth nature necessitates a subband decomposition which contains an approximation signal, for which a sparsity prior is ill-suited.

To address these assumptions, we designate the first channel of the sparse code as the approximation signal and fix its corresponding synthesis filter to an analytic low-pass filter. Note that total variation regularization of the low-pass signal has been previously studied (Elad et al., 2005; Lalanne et al., 2020), however, we're concerned with regularizing the dictionary elements for reasons concerning mutual coherence. Knowing in which subband the approximation signal lives allows us to remove it from the soft-thresholding operation ($\theta_0^{(k)} = 0$), thereby removing any misplaced assumption of sparsity. Further, we wish to ensure no additional low-pass filters are learned during training so that we are not inadvertently violating the sparsity assumptions of the model (i.e. thresholding other low-frequency subbands) and reduce the mutual coherence of dictionary. This restriction on the number of low-pass filters has the added benefit of improving stable recovery bounds of the dictionary as discussed in Section 2.1.1.

The issue of learning high-pass/band-pass filters is both non-trivial and ill-posed. If we naively assert that such a set of filters must simply be "non-low-pass", we may consider projecting filters onto the set of zero-mean filters there by removing their DC-component. However, this allows for the learning of filters whose frequency response is arbitrarily close to DC. Alternatively, preprocessing the signal by removing its low-pass component is not effective as it produces a noisy low-frequency signal and does not properly regularize the learned filters. As demonstrated in Appendix 5.1, even when the input images are preprocessed in this way, the learned dictionary can still contain low-pass filters leading to high mutual coherence.

A more apt characterization is to consider the learning of filters occupying the frequency-space complement to that of the low-pass filter. Let $\boldsymbol{h}$ denote a fixed low-pass filter and $\boldsymbol{g} = \boldsymbol{\delta} - \boldsymbol{h}$ be its high-pass complement, where $\boldsymbol{\delta}$ is the discrete Dirac delta function. We formalize the regularization by considering the following effective dictionary elements,

$$\boldsymbol{d}^1 = \boldsymbol{h}, \quad \boldsymbol{d}^j = \boldsymbol{g} * \tilde{\boldsymbol{d}}^j, \quad \|\boldsymbol{d}^j\|_2 \leq 1, \quad j = 2, \ldots, M. \tag{7}$$

We refer to $\{\boldsymbol{d}^j\}_{j=1}^M$ and $\{\tilde{\boldsymbol{d}}^j\}_{j=2}^M$ as the effective and learned filters respectively. Signal reconstruction is ultimately performed with the effective filters which compose $\boldsymbol{D}$. Note that the norm constraint is necessary to avoid large responses in the transition band of the low-pass filter. By explicitly denoting which subbands of our decomposition are expected to be sparse, this regularization technique forms a sufficiently expressive model for the reconstruction of natural images. This has the added benefit of nearly eliminating the correlation between the atoms corresponding to the lowpass filter, $\boldsymbol{d}^1$, and the atoms corresponding to high-frequency filters, $\boldsymbol{d}^j$, as $\boldsymbol{d}^1 * \boldsymbol{d}^j = \boldsymbol{h} * (\boldsymbol{\delta} - \boldsymbol{h}) * \tilde{\boldsymbol{d}}^j = (\boldsymbol{h} - \boldsymbol{h} * \boldsymbol{h}) * \tilde{\boldsymbol{d}}^j \approx 0$, for $j \neq 1$.

## 2.3 BLIND DENOISING: NOISE-ADAPTIVE LEARNED THRESHOLDS

As presented, the CDLNet model and any similar network utilizing LISTA is not amenable to generalizing denoising performance across a set of noise levels. Note that the threshold values in soft-thresholding operator are directly proportional to the expected sparsity and the noise level in each subband Bayram & Selesnick (2010). As a result, the sparsity hyperparameter, $\lambda$, and consequently the threshold values should be functions of the noise variance, i.e. $\theta^{(k)} = \theta^{(k)}(\sigma_n^2)$.

We thus propose to parameterize the thresholds in the last layer in CDLNet as $\boldsymbol{\theta}^{(K)} = \boldsymbol{\nu}^{(K)}\hat{\sigma}_n^2$, where $\hat{\sigma}_n^2$ is the estimated noise variance which can be estimated from the input noisy image, and $\boldsymbol{\nu}^{(K)}$ is a vector containing the learned scaling factors for different subbands. We employ a commonly used estimator, $\hat{\sigma}_n \approx \mathrm{Median}(|c|)/0.6745$, where $c$ denotes the diagonal-detail Wavelet subband of an input image (Chang et al., 2000; Mallat, 2008; Donoho & Johnstone, 1994; 1995). The proposed parmeterization of thresholds is inspired by the MAP estimate of orthogonal Wavelet denoising under a Laplace distribution prior of the high-frequency coefficients and the Gaussian distribution prior on the noise(Bayram & Selesnick, 2010). This parameterization enables the proposed CDLNet to handle varying input noise-levels while maintaining the integrity of CDLNet as an unfolded dictionary learning model.

## 3 EXPERIMENTAL SETUPS AND RESULTS

**Models:** are trained via stochastic gradient descent on the $\ell_2$-loss with parameter constraints,

$$\underset{\Theta=\{\{\boldsymbol{A}^{(k)},\boldsymbol{B}^{(k)},\boldsymbol{\theta}^{(k)}\}_{k=0}^{K-1},\{\boldsymbol{d}^j\}_{j=1}^M\}}{\mathrm{minimize}} \|\boldsymbol{x} - \hat{\boldsymbol{x}}(\boldsymbol{y};\Theta)\|_2^2 \quad \text{subject to:} \quad \boldsymbol{\theta}^{(k)} \geq 0 \quad \forall k, \quad \|\boldsymbol{d}^j\|_2^2 \leq 1 \quad \forall j,$$

where the parameter constraints are enforced by projection onto the constraint set after each gradient step. Models of different capacity are trained by varying the number of unrollings $K$ and number of subbands $M$. Filters are of size $7 \times 7$. CDLNet is used to refer to our proposed base-model, differing from other mentioned CDL methods (ACSC (Sreter & Giryes, 2018) and CSCNet (Simon & Elad, 2019)) by its use of stride-2 convolutions, mean-subtraction of input signals, and the above projection operations during training. A $3 \times 3$ isotropic Gaussian filter ($\sigma$=0.6) is used as the analytic low-pass filter for frequency-regularized models, denoted FCDLNet. We use (F)CDLNet+Blind to refer to networks with noise-adaptive thresholds as in section 2.3. In blind denoising cases, the noise level is estimated using the estimator in section 2.3 both during training and inference. Implementation and trained models are provided *here*[1].

**Dataset:** All CDLNet models and variants are trained on the BSD432 dataset (Martin et al., 2001). Random crops of size $128 \times 128$ are flipped, rotated, and batched online during training. Independent identically distributed Gaussian noise is drawn from $\sigma_n \in \sigma_n^{\mathrm{train}}$ uniformly within each batch and added to the ground-truth signal. As preprocessing, all images are normalized by 255 to have range of $[0, 1]$ and mean of each image is subtracted. Testing is performed on the associated BSD68 test-set (Martin et al., 2001).

**Training:** is performed with the Adam optimizer (Kingma & Ba, 2015), using its default settings in PyTorch. Mini-batches consist of 10 samples. A learning rate of `1e-3` is set at the start of training and reduced by a factor of 0.95 every 50 epochs. Training is run until convergence. As advised by Lecouat et al. (2020), backtracking is used to correct for model divergence by reloading the most recent checkpoint within the last 10 epochs and reducing the learning rate by a factor of 0.8.

**Initialization:** A single set of $M$ filters are initialized by drawing from a standard normal distribution and subsequently normalized w.r.t each filter. This corresponds to our expectation that most filters will learn to be approximately zero-mean and spatially localized. We found that this initialization greatly improves convergence speed over drawing from a standard uniform distribution. All convolution operators are initialized with this same weight. Following Simon & Elad (2019), we then normalize $\boldsymbol{A}^{(k)}$ by the spectral norm $L = \|\boldsymbol{A}^{(k)}\boldsymbol{B}^{(k)}\|_2$, which corresponds to initializing the step-sizes of ISTA to $\eta^{(k)} = 1/L$. Thresholds are initialized to $\boldsymbol{\theta}^{(k)} = $ `1e-1`$/L$.

### 3.1 EFFECT OF FREQUENCY REGULARIZATION AND STRIDE ON LEARNED DICTIONARIES

To validate the effectiveness of small-stride and the proposed frequency regularization on the learned synthesis dictionary, we train three CDLNet models containing convolutions with (a) no stride, (b) stride 2, and (c) stride 2 with frequency regularization. For all models $M$=32, $K$=20, and $\sigma_n^{\mathrm{train}}$=25. Figure 2 shows the learned filters in the spatial and frequency domain. Without stride, the learned dictionary consists of some "noise-like" filters with non-localized frequency responses and a few directional filters. The stride 2 model (b) learns more directional filters and overall a dictionary with

---

[1]https://www.dropbox.com/sh/6lgy1w5v6b5b4jc/AAC8Cwgshu0h8ySfwivSRZpMa?dl=0

lower mutual-coherence compared to (a). However, both (a) and (b) produce multiple low-frequency filters in unpredictable channels. With frequency regularization added in (c), we are able to control the subband in which our low-frequency information is located. The learned filters in (c) are all directional or texture high-pass, and the mutual-coherence is decreased as predicted.

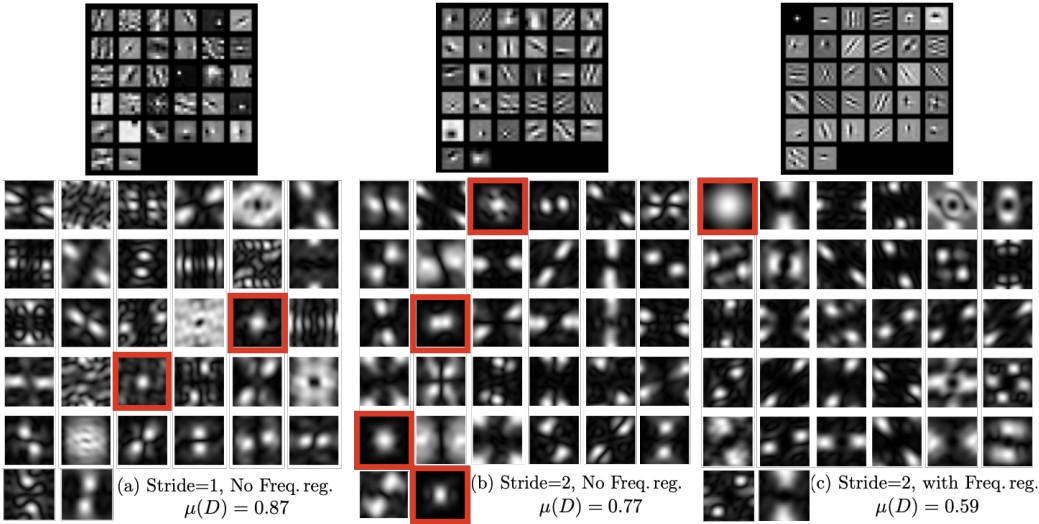

(a) Stride=1, No Freq. reg.
$\mu(D) = 0.87$

(b) Stride=2, No Freq. reg.
$\mu(D) = 0.77$

(c) Stride=2, with Freq. reg.
$\mu(D) = 0.59$

Figure 2: Learned Filters (top row) and their corresponding frequency responses (bottom row). Boxes highlight low-pass filters as seen in the frequency domain. Observe that non-frequency regularized dictionaries (a,b) have multiple of such filters in non-determined locations, in contrast to (c). The mutual coherence between dictionary elements ($\mu(\boldsymbol{D})$) is reduced by incorporating stride and frequency regularization. Filters in (c) are more interpretable than those reported in ACSC (Sreter & Giryes, 2018) and CSCNet (Simon & Elad, 2019)

## 3.2 Denoising performance against other frameworks

In this section we demonstrate the efficacy of the proposed methods on single noise-level grayscale image denoising. We train two FCDLNet models of varying capacity (FCDLNet with $M$=64, $K$=10 and Big FCDLNet with $M$=169 and $K$=30)[2]. We compare these to the classic collaborative filtering method BM3D (Dabov et al., 2007), popular convolutional neural network based methods FFDNet (Zhang et al., 2018) and DnCNN (Zhang et al., 2017), and CDL method proposed by Simon & Elad (2019), CSCNet. All learned methods have been trained on the same dataset, BSD432. Average peak signal-to-noise ratio (PSNR) on BSD68 testset is shown in Table 1. Visual comparison between the above mentioned models and FCDLNet is presented in Figure 3.

The FCDLNet with trainable parameters on the order of CSCNet shows improved performance across noise-levels. Interestingly, Big FCDLNet is observed to compete very well with state-of-the-art deep-learning denoising networks. This is done without the use of common deep-learning tricks such as batch-normalization or residual learning (both of which are employed in DnCNN). The ability to train larger CDLNet models of competitive performance without such methods may suggest an appeal to more interpretable networks.

The average run-time at inference of different models averaged over Set-12 (Sreter & Giryes, 2018) images of size $512 \times 512$ is also given in Table 1. The timing experiments were conducted with an Intel Xeon E5 at 2.6GHz CPU, an Nvidia P40 GPU, and 4GB of RAM, running Linux version 3.10.0. We observe that by leveraging small-strided convolutions and forgoing the "shift-duplicate processing" of CSCNet, FCDLNet has significantly reduced (10x to 20x) computation time both on GPU and CPU compared to CSCNet, while having better denoising quality.

---

[2]Corresponding filters are available here.

Table 1: Denoising performance (PSNR) on BSD68 testset ($\sigma = \sigma_n^{\text{train}} = \sigma_n^{\text{test}}$).

| $\sigma$ | BM3D | FFDNet | DnCNN | CSCNet | FCDLNet | Big FCDLNet |
|---|---|---|---|---|---|---|
| 15 | 31.07 | 31.63 | **31.72** | 31.40 | 31.45 | 31.66 |
| 25 | 28.57 | 29.19 | **29.22** | 28.93 | 28.99 | **29.22** |
| 50 | 25.62 | 26.29 | 26.23 | 26.04 | 26.11 | **26.30** |
| Params | - | 486k | 556k | 64k | 66k | 510k |
| CPU time (sec) | 17.06 | - | - | 14.76 | 0.76 | 9.93 |
| GPU time (sec) | - | - | - | 0.34 | 0.03 | 0.14 |

(a) Original    (b) Noisy (20.13 dB)    (c) BM3D (28.68 dB)    (d) CSCNet (29.34 dB) (e) FCDLNet (29.38 dB)

Figure 3: Visual comparison of different models for noise level $\sigma_n = 25$. PSNR value for each image is given in parentheses. Details are better visible by zooming on images.

### 3.3 ROBUSTNESS TO NOISE LEVEL MISMATCH IN TRAINING AND INFERENCE

In this section we provide experimental results regarding the generalization of the networks across noise-levels. The main focus is to investigate the effect of the proposed blind denoising framework (section 2.3), especially for cases with mismatch between noise-range during training ($\sigma_n^{\text{train}}$) and testing ($\sigma_n^{\text{test}}$).

In Figure 4 we show the average PSNR values for three different training noise ranges: (a) $[0, 20]$, (b) $[15, 35]$, and (c) $[30, 50]$. Networks are trained by uniformly sampling the noise-level within the training range at each iteration. All networks have close to 120k learnable parameters with $M$=64 and $K$=20. The trained networks are then tested on different noise levels $\sigma_n^{\text{test}} = [0, 50]$, and average PSNR is calculated over the BSD68 dataset.

As shown in Figure 4, all networks perform closely over the training noise-range. On the other hand, when tested on noise-levels outside the training range, the networks with adaptive thresholds (as in Section 2.3) perform superior compared to others. In spite of increasing input signal-to-noise ratio for noise-levels below the training range, we observe that models without noise-adaptive thresholds have diminishing performance returns (note the plateau of CDLNet/FCDLNet in $\sigma_n^{\text{test}} = [0, 15]$ in (b) and $\sigma_n^{\text{test}} = [0, 30]$ in (c)). On the other hand, denoising behavior of models with noise-adaptive thresholds (CDLNet+Blind and FCDLNet+Blind) extends to the lower noise-range. Similarly, we observe that models without noise-adaptive thresholds have a more significant performance drop compared to noise-adaptive models when generalizing above the training noise level. Another notable observation is that FCDLNet models perform better than their non-frequency regularized counterparts in low noise-levels due to the proper treatment of the low-pass signal.

We also compare the generalization of the proposed networks against other CDL methods. Pokala et al. (2020) propose ConFirmNet model where they use firm-thresholding in LISTA and show better performance compared to ACSC (Sreter & Giryes, 2018) when training and testing noise levels are different. Results from Pokala et al. (2020) are summarized and compared to our framework in Table 2. FCDLNet performs on par with ConFirmNet when $\sigma_n^{\text{train}} = 20$. To allow the proposed scaling parameters ($\boldsymbol{\nu}^{(K)}$) to properly fit to the noise-variance, we train over $\sigma_n^{\text{train}} = [18, 22]$. As seen in Table 2 and from our discussion above, simply training over a noise range gives marginal improvement. However, when combined with noise-adaptive thresholds (FCDLNet+Blind), we observe significant improvement in generalization over other methods.

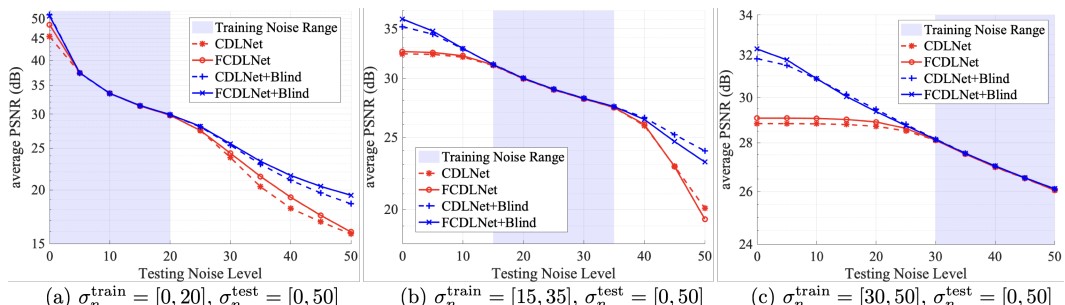

Figure 4: Generalization of the network for different training noise range. Average PSNR is calculated over BSD68 testset.

Table 2: Generalization of the network: mean (sd) PSNR on Set-9 (Pokala et al., 2020) testing set. Comparisons with ACSC (Sreter & Giryes, 2018) and ConFirmNet (Pokala et al., 2020).

| $\sigma_n^{\text{test}}$ | $\sigma_n^{\text{train}} = 20$ | | | | $\sigma_n^{\text{train}} = [18, 22]$ | | |
|---|---|---|---|---|---|---|---|
| | ACSC | ConFirmNet | CDLNet | FCDLNet | CDLNet | FCDLNet | FCDLNet + Blind |
| 5 | 32.02 (0.02) | 32.23 (0.01) | 32.04 (0.01) | 32.17 (0.01) | 32.76 (0.01) | 32.81 (0.01) | **34.25** (0.02) |
| 15 | 31.88 (0.03) | 32.04 (0.03) | 32.00 (0.03) | 32.06 (0.03) | 32.24 (0.03) | 32.30 (0.04) | **32.45** (0.03) |
| 30 | 22.89 (0.03) | 23.13 (0.04) | 23.68 (0.04) | 23.70 (0.05) | 24.34 (0.06) | 24.51 (0.05) | **25.31** (0.06) |

## 4 DISCUSSION AND CONCLUSION

In this study we investigated unrolled convolutional sparse coding and dictionary learning frameworks. These frameworks have the benefit of interpretability while maintaining similar performance compared to other state-of-the-art deep learning models. We proposed employing a strided convolutional dictionary constructed with a fixed lowpass filter and a set of learned frequency regularized filters. As illustrated, small-strided and frequency regularized convolutions give the benefit of reduced mutual coherence of the dictionary and properly address the modeling assumptions regarding convolutional sparse coding. We showed that learned high-pass filters are more structured covering different orientations and textures. In comparison to other CDL models of similar parameter count, our proposed framework showed improved denoising performance whilst reducing the computational cost. The learned dictionary filters are more interpretable with lower mutual coherence. Additionally, experimental results with FCDLNet models of similar size to the state-of-the-art denoising models showed competitive denoising performance.

We further investigated the generalizability of CDL networks in scenarios where noise-level mismatch exists between training and inference. Leveraging the interpretability of CDLNet, we proposed to parameterize the thresholds in LISTA such that they are scaled based on the estimated input noise variance. Experimental results demonstrated that this reparameterization greatly improves the robustness to noise-level mismatch between training and testing and increases the generalizability of the network.

In future work we aim to explore the possible extensions of the proposed models and further leverage the interpretability of this framework. The proposed frequency regularization scheme provides the required grounds for multiresolution representation learning. Note that by further processing of the fixed lowpass channel one can achieve a multiresolution representation while in other frameworks the lowpass information is represented in multiple, non-predetermined channels, making this extension challenging (see discussion in Section 3.1). Further augmenting the thresholds of the CDLNet model to be employed at each layer of LISTA, with both signal and noise adaptivity, is a promising direction for improved generalization of the network. Additionally, investigating other noise distribution models is an exciting avenue of research.

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

## 5 APPENDIX

### 5.1 REMOVING THE LOW-PASS INFORMATION FROM THE INPUT IMAGES IS NOT EFFECTIVE FOR REDUCING THE MUTUAL COHERENCE

Instead of the proposed frequency regularization approach, it may be tempting to simply remove the low-pass information of the signal as a preprocessing step. More specifically, let $y$ be the noisy image, one can process the high-pass signal $y - h * y$ with the proposed network without adding any frequency regularization. Note that this approach is not equivalent to the proposed frequency regularization scheme as the removed low-pass channel ($h * y$) is still noisy. Although the thresholds for the low-pass channel in FCDLNet are set to zero, this does not mean that the low-pass information is removed from the denoising framework. As a result, in FCDLNet, the low-pass filtering at each LISTA stage is ultimately a filtering of an incrementally cleaner image, producing a full subband decomposition of a cleaned up low and high frequency bands at the final stage of LISTA. Additionally, note that removing the low-frequency component of the signal does not stop the learned dictionary from learning multiple (redundant) low-pass filters. Even though the input signal does not contain low-frequency information, the filters are not necessarily regularized to take advantage of this property. As shown in figure 5, we observed that the filters learned in this scheme have multiple low-pass channels.

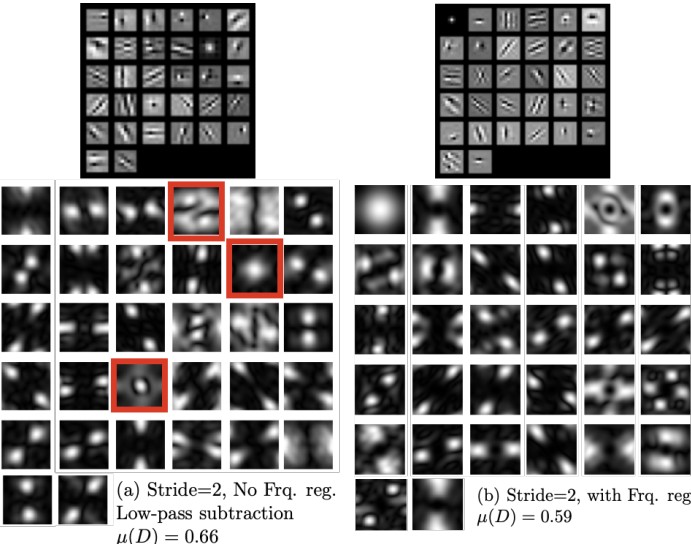

Figure 5: Learned Filters (top row) and their corresponding frequency responses (bottom row). Boxes highlight low-pass filters as seen in the frequency domain. Observe that non-frequency regularized dictionary when low-pass information is removed (a) has multiple of such filters in non-determined locations, in contrast to (b). The mutual coherence between dictionary elements ($\mu(\boldsymbol{D})$) is reduced by incorporating the frequency regularization.

