# OpenReview forum: "Frequency Regularized Deep Convolutional Dictionary Learning and Application to Blind Denoising"
_ICLR.cc/2021/Conference — Reject_

### Official Review · AnonReviewer3 · 2020-10-28
**Limited novelty.**

**Rating:** 4
**Confidence:** 5

**Review:**

This paper proposed a convolutional dictionary learning (CDL) work for image denoising. Compared with existing CDLnet, the proposed method used stride convolutions with pre-defined low-pass channels  to improve the performance and reduce complexity. Results on image denoising showed comparable performance to the state-of-the-arts.


Strength:

1. The idea of constraining low-pass atoms in learned dictionary is interesting, which limits the space of learned atoms to have the desired behavior.

2. The proposed method parameterize the soft-thresholding operator in LISTA such that the thresholding is directly linked with the estimated image noise level.

Weaknesses:

1. The contribution of this paper is incremental. The proposed method is heavily based on existing CDLNet with some modifications. Although such modifications improved results a little bit, the motivation is not very clear and the implementation is empirical, e.g., Eq. (7).

2. The proposed method did not substantially improve the results. It improves the baseline CSCNet a little bit, but still not good as the state-of-the-art methods such as DnCNN. In addition, all the experiments are using synthetic data with ideal Gaussian noise. Actually image noise is more complex and is a mixture of Poisson and Gaussian. Under controlled environment, the proposed method still cannot outperform the state-of-the-art. The results on real image noise might be even worse.

Abdelhamed, Abdelrahman, Stephen Lin, and Michael S. Brown. "A high-quality denoising dataset for smartphone cameras." Proceedings of the IEEE Conference on Computer Vision and Pattern Recognition. 2018.

Plotz, Tobias, and Stefan Roth. "Benchmarking denoising algorithms with real photographs." Proceedings of the IEEE conference on computer vision and pattern recognition. 2017.

3. More analysis of the proposed components are missing. For example, there is no ablation study for the proposed components.

---

> ### Author Response · Authors · 2020-11-25
> **Reviewer 3 Response**
>
> The authors would like to thank the reviewer for taking the time and careful reading of the manuscript and providing insightful comments that help improving the manuscript. Please see as follows how we have addressed the comments.
>
> 1. We have improved the introduction to better present the motivation of the paper. Our proposed baseline network (CDLNet) differs from the original framework of [Sreter2018], ACSC, by including mean-subtraction preprocessing, small-strided convolutions, and a norm-constraint on the dictionary. Our proposed frequency regularization method provides an alternative strategy for addressing the problem of mutual coherence as proposed by [Simon2019] (CSCNet). Our framework is both more computationally efficient and more consistent with the convolutional model (not the patch-averaging). Furthermore, our Big FCDLNet demonstrates truly competitive performance with state-of-the-art deep-learning networks, which cannot be said for either the ACSC or CSCNet models. Lastly, we show that the interpretability of the network can be leveraged to propose a blind-denoising method which outperforms that of [Sreter2018,Pokala2020]. Regarding the comment about empirical implementation, please  note that the formulated frequency regularization presented in equation 7 is a soft-constraint on the energy of the atoms in the frequency domain, based on convolution theorem. It allows the learned filters to not be over-constrained while still introducing the desired low-high frequency separation.
>
> 2. Please note that one of the focus points of the study is the learned dictionary. More specifically, we introduce an alternative approach to CSCNet when addressing the mutual coherence issue of the learned dictionary. As explained in the manuscript, the proposed framework utilizes a frequency regularization scheme and a small-stride to overcome the coherence issue of the dictionary. The CSCNet framework uses a large stride and shift and averaging scheme which is computationally expensive. In terms of denoising performance, we have shown improvement compared to CSCNet while reducing the inference time (as presented in Table I). Additionally, as [the attached image](https://www.dropbox.com/s/swhxu9k22cm2hfh/acsccscnetfcdl.png?dl=0) shows, the learned filters using the FCDLNet framework have more Gabor or texture like filters. Whereas, the filters learned in ACSC or CSCNet frameworks lack such structures.
>
> Note that the proposed framework achieves comparable denoising performance (-0.2 dB) compared to other deep-learning frameworks (DnCNN and FFDNet) with significantly less number of parameters, without using common tricks such as batch-normalization and with a more interpretable structure. Also note that although CSCNet introduces an interpretable framework, the authors do not explore the performance as the number of parameters increases. In this study, we further show that by increasing the number of parameters to similar values as of the other deep-learning methods (e.g., DnCNN and FFDNet), we can get comparable results (shown in Table I). This further shows that the common deep-learning tricks are not the only way to achieving such performance and other, interpretable, structures can also achieve similar results.
>
> In terms of the performance with real-world noise, in this study we have shown comparisons on additive white Gaussian noise which is an extremely common practice in the literature. Note that all the benchmark papers are also using the same experimental setup. The manuscript introduces a framework and we believe presented experiments are adequate for supporting the benefits of using frequency regularization, and for the proposed noise-adaptative parameterization of the soft threshold in the paper. The performance of the proposed method on other noise distributions can be an interesting avenue for future study. For readers interested in other noise distributions we have cited the mentioned papers in the introduction.
>
> 3. Please note that the main components of the proposed framework are: frequency regularization, small stride, and the noise-adaptive thresholds. The effect of the stride and frequency regularization have already been studied in section 3.1. In section 3.3 the effect of adding the noise-adaptive thresholds has been shown.

---

### Official Review · AnonReviewer2 · 2020-11-01
**Subpar in terms of significance and clarity, needs better development**

**Rating:** 3
**Confidence:** 3

**Review:**

The paper proposes a new regularization for the dictionary in the learned convolutional sparse coding model of Sreter & Giryes '18. The main contribution is that the dictionary is regularized to be composed of 1) a fixed low-pass filter and 2) a set of learned filters to occupy the complementary high-frequency space. A second contribution is that the thresholding in the network is adjustable according to the estimated noise level in the image.

Comments:

- Presentation-wise, I didn't get the motivation of the work after reading the introduction. Everything before Sec. 1.2 is a narrative of the existing methods, and all of a sudden Sec. 1.2 states the proposed method, but what is the problem that the paper wants to address?

- The major contribution is the introduction of a fixed low-pass filter in the dictionary, but I don't see a clear justification as to why it is needed. If you want to model the DC component, why not simply use a bias term?

- The paper tells a story that the dictionary need to be incoherent, that some prior work enforces this by using large stride convolution, and that the proposed method does not need to use large stride. But ultimately there is no incoherence regularization for the high-frequency part of the dictionary in the proposed method. So how do you enforce those high-frequency filters to be incoherent?

- It is stated as a second main contribution of the paper that the thresholding in the network can be made to be dependent on the noise level in the image, so that the network is capable of performing blind denoising. However, the explanation for how it works (Sec. 2.3) is very sketchy. Why is the adaptive threshold only used in the last layer? Also, the threshold is set to be a multiple of noise variance, but how do you estimate the noise variance in practice? Where does the number 0.6745 come from?

Overall, the presentation of the paper needs major improvement to make the motivation as well as technical details clear. In addition, the technical contributions appear quite minor and are not fully justified: it is unclear how fixed low-pass filter compares with a bias term, and it is unclear how effective the adaptive threshold is in practice.

---

> ### Author Response · Authors · 2020-11-25
> **Reviewer 2 Response**
>
> The authors would like to thank the reviewer for taking the time and careful reading of the manuscript and providing insightful comments that help improving the manuscript. Please see as follows how we have addressed the comments.
> 1. We have modified the introduction section to better introduce and motivate the problem. We hope the reviewer agrees that it is more clear now. This work focuses on learning convolutional dictionaries for image representation through the use of the Learned ISTA (LISTA) network for sparse coding. We aim to address two outstanding issues in this framework: 1) to reduce the mutual coherence of the learned dictionary, and 2) to learn a single network that can work effectively over a large range of the noise level.
> 2. As discussed in Section 2.1.1, the piecewise-smooth nature of natural images necessitates that a dictionary must contain a low-frequency signal. Although a constant (i.e. a DC component) is a low-frequency signal, it may not be the best low-frequency signal that leads to the most efficient signal representation. In fact, most well-established orthogonal dictionaries (e.g. wavelets or DCT) do not use a constant atom to represent the low-frequency signal. Furthermore, a problem arises in using a decomposition with multiple low-pass filters: low-pass filters and their translates will lead to significantly high mutual coherence (by their smooth varying nature). We thus choose to model the low-frequency component with a fixed low-pass filter, and constrain the remaining filters to exists in the complementary frequency space (with a soft-constraint enforced by convolution theorem, eqn. 7). This restricts the learned filters from occupying the frequency-domain arbitrarily close to DC, producing no learned "smoothly-varying" filters, as verified experimentally in Figure 2. To further reduce the mutual coherence (caused by the single low-pass filter) we proposed to use a significantly small stride (stride=2). This will effectively reduce the mutual coherence without the need for shift and average processing required by the large stride presented in [Simon2019].
> 3. As shown in Fig. 1 in the manuscript, the learnt high-pass filters naturally reside in different parts of the frequency spectrum, with low mutual coherence between them. This is achieved without explicitly penalizing the coherence among the filters. The main source of the mutual coherence in the dictionaries learnt with prior methods  is the presence of multiple low pass filters. By constraining the dictionary to have only a single low-pass filter, with the rest being high-pass filters in the complimentary part of the frequency spectrum of the low-pass filter, our learned filters have much lower mutual coherence than the prior work. In our future work, we will consider how to further constrain the high-pass filters to minimize the mutual coherence.
> 4. The adaptive threshold is employed in the last layer only based on experimental observation. One possible explanation for this phenomenon is that the learned adaptive thresholds are noise-dependent, and not signal-dependent. Exploring this is an interesting area for future work and has been added to the conclusion of the paper. We believe that the proposed method demonstrates the ability to leverage the interpretability of the network to propose improvements based on classical signal processing knowledge. As explained in Section 2.3, we employ the popular robust estimator based on the diagonal coefficients of a wavelet decomposition of the noisy image. We refer the reviewer to the already mentioned references provided for further details on the estimators introduction [Donoho1994] [Donoho1995] [Mallat2008] [Chang2000]. The number $0.6745$ is derived from the median of a half-normal distribution, $\sigma \sqrt{2}\mathrm{erf}^{-1}(1/2)$.

---

### Official Review · AnonReviewer5 · 2020-11-05
**The method seems to be a small extension of CDLNet and the evaluation could be improved.**

**Rating:** 4
**Confidence:** 4

**Review:**

## Summary

The paper proposes a denoising method with a neural network inspired from convolutional dictionary learning. In the proposed method, one atom of the dictionary is constrained to be a low frequency filters and all other atoms are to be high frequency atoms. The authors also propose to make the threshold depends on the noise level to better adapt to different noise level and to use strided convolution to reduce the computational cost of the method. The method is then evaluated on images from BSD68.

----

*For extra citations, see bibtex at the end. Sorry if I ended up putting a lot of them but I feel the bibliography is a bit lacking.*

## Overall assessment

- A major weakness of this work is that it seems to be equivalent to doing CDL on a signal filtered with high pass filter $g^{-1} * y_i$ because $\theta_0 = 0$. Indeed, using parseval on the data fitting term and the equivalence between convolution and point wise multiplication in the frequency space, one can easily show that $z^1_i$ will correspond to the low frequency part of $y_i$ and as the filters $d^i$ all only have high frequency, they will try to reconstruct the high frequency part of $y_i$. This means that only keeping high frequency of $y_i$, one can use the classical CDL approach algorithm to recover the same method as the one proposed in this paper. To me, the method boils down to stating an equivalent model where the preprocessing is integrated in the model. If the authors think I am wrong, they could try to perform experiment 3.1 and show that there is significant difference between filtering $y$ and using the proposed method. At least, this point should be mentionned and discussed in the manuscript. Note that such approach of integrating multiple component can be related to the work on morphological component analysis (see `[Elad2005]`) and the integration of preprocessing step in CDL have recently been proposed with detrending in `[Lalanne2020]`.
- Apart from this point, the novelty of this work is not very significant. The adaptation of the thresholds with the input noise level in the context of denoising has been proposed in `[Isogawa2017]` and `[Ramzi2020]` and  the use of strided convolution are proposed in `Simon & Elad (2019)`.
- The effect of the stride on the denoising is not evaluated. This would be interesting to look at how the denoising performance change when changing the stride. In particular, how does FCDLNet compare with FCDLNet with `stride` $\in\\{1, 3, 4\\}$. Does it impact the performances a lot?
- It would also be interesting to study the impact of the noise level estimator. If the estimator is biased, how does it impact the performances? This is also related to Question 3, as I suppose if the estimator is biased but used for training, the thresholds $\nu$ can also be adapted to cope for this bias. This would be interesting to add such experiments.
- For the computational complexity, I would be interested to see comparison with modern convolutional sparse coding algorithm such as the one in `sporco` (`[Wohlberg2017]`) or the LGCD algorithm from `[Moreau2019]`, which scales to much larger images.
- The writing is not very clear and not always correct and there are many typos (see bellow).


## Some question

1. Could the authors comment on why the learned network is more interpretable than a classical network?
2. Why  does the authors change the constraint to the one in (7) ? This makes the model somewhat incomparable with other approaches as $d^j$ won't have the unit norm property. Moreover, I don't really see the point, as simply stating that $\\|d^j\\| \le 1$ also constrains appropriatly $\\|\tilde d^j\\|$ to be smaller than roughly $\frac{1}{\\|g\\|}$, which is a similar constraint but comparable to the original one.
3. When training the network, is the true noise level given as an input of the model or is it also estimated using the wavelet based estimator? This is unclear from the text and should be clarified.
4. What is the training time for the proposed model? This is not discussed, but I guess this is similar to the training time of other models.


## Minor comments, nitpicks and typos

- For the citation, when they are between parenthesis, could the authors use `\citep` to have proper formating.
- p.1: `Mairal et al. (2014)`: This is not an arXiv paper, the proper citation is `[Mairal2014a]`.
- p.1: The first paragraph could be improved a lot. It is unclear why this would be an inverse problem (there is no sensing matrix here, the dictionary correspond to the prior knowldge in the inverse problem literature). This is mainly denoising so the paragraph should be fixed to reflect this.
- p.1: `a linear combination of a collection of vectors`: this is simply linear representation. The sparse linear representation also promotes the usage of only a few atoms.
- p.1: `where $n_i \sim \mathcal N(0, \sigma^2 \bm I)$.` The authors should add in plain word that this `is an additive Gaussian white noise.`
- p.2: `is nontrivialy` -> `non trivialy`? What does it mean to be trivialy related? I would remove this as it is unclear.
- p.2: `the Convolutional Sparse Coding (CSC) model has been introduced`: the original work introducing such model is `[Grosse2007]`.
- p.3: `interpretabile` -> `interpretable`
- p.4: Note that there is a third option for training LISTA networks which is to use loss in Eq.(5) as a training loss, as it is done for instance in `[Ablin2019]`.
- p.5: `the Guassian distribution prior` -> `Guassian`
- Table.1: Could the authors highlight the leading method in the table?
- p.7: It is unclear whether the timing for FCDLNet is performed for full image denoising or on 128x128 patches.
- Figure.4: Could the authors add the error bars in this plot? As the gain is small between CDLNet and FCDLNet, it would be interesting to see the confidence here.

### References

```bibtex
@inproceedings{Ablin2019,
  title = {Learning Step Sizes for Unfolded Sparse Coding},
  booktitle = {Advances in {{Neural Information Processing Systems}} ({{NeurIPS}})},
  author = {Ablin, Pierre and Moreau, Thomas and Massias, Mathurin and Gramfort, Alexandre},
  year = {2019},
  pages = {13100--13110},
  address = {{Vancouver, BC, Canada}},
  archivePrefix = {arXiv},
  copyright = {All rights reserved},
  eprint = {1905.11071},
  eprinttype = {arxiv}
}

@article{Elad2005,
  title = {Simultaneous Cartoon and Texture Image Inpainting Using Morphological Component Analysis ({{MCA}})},
  author = {Elad, Michael and Starck, J. L. and Querre, P. and Donoho, D. L.},
  year = {2005},
  volume = {19},
  pages = {340--358},
  journal = {Applied and Computational Harmonic Analysis},
  number = {3},
  pmid = {16370462}
}

@article{Grosse2007,
  title = {Shift-{{Invariant Sparse Coding}} for {{Audio Classification}}},
  author = {Grosse, Roger and Raina, Rajat and Kwong, Helen and Ng, Andrew Y.},
  year = {2007},
  volume = {8},
  pages = {9},
  journal = {Cortex}
}

@article{Isogawa2017,
  title={Deep shrinkage convolutional neural network for adaptive noise reduction},
  author={Isogawa, Kenzo and Ida, Takashi and Shiodera, Taichiro and Takeguchi, Tomoyuki},
  journal={IEEE Signal Processing Letters},
  volume={25},
  number={2},
  pages={224--228},
  year={2017},
  publisher={IEEE}
}

@inproceedings{Lalanne2020,
  title = {Extraction of {{Nystagmus Patterns}} from {{Eye}}-{{Tracker Data}} with {{Convolutional Sparse Coding}}},
  booktitle = {2020 42nd {{Annual International Conference}} of the {{IEEE Engineering}} in {{Medicine}} \& {{Biology Society}} ({{EMBC}})},
  author = {Lalanne, Clement and Rateaux, Maxence and Oudre, Laurent and Robert, Matthieu P. and Moreau, Thomas},
  year = {2020},
  month = jul,
  pages = {928--931},
  publisher = {{IEEE}},
  address = {{Montreal, QC, Canada}},
}

@article{Mairal2014a,
  title = {Sparse {{Modeling}} for {{Image}} and {{Vision Processing}}},
  author = {Mairal, Julien and Bach, Francis and Ponce, Jean},
  year = {2014},
  volume = {8},
  pages = {85--283},
  journal = {Foundations and Trends\textregistered{} in Computer Graphics and Vision},
  number = {2-3}
}

@article{Moreau2019,
  title={Distributed Convolutional Dictionary Learning (DiCoDiLe): Pattern Discovery in Large Images and Signals},
  author={Moreau, Thomas and Gramfort, Alexandre},
  journal={arXiv preprint arXiv:1901.09235},
  year={2019}
}

@inproceedings{Ramzi2020,
  title = {Wavelets in the {{Deep Learning Era}}},
  booktitle = {European {{Signal Processing Conference}} ({{EUSIPCO}})},
  author = {Ramzi, Zaccharie and Starck, Jean-Luc and Moreau, Thomas and Ciuciu, Philippe},
  year = {2020},
  month = jul,
  pages = {1417--1421},
}

@inproceedings{Tolooshams2018,
  title = {Scalable Convolutional Dictionary Learning with Constrained Recurrent Sparse Auto-Encoders},
  booktitle = {{{IEEE International Workshop}} on {{Machine Learning}} for {{Signal Processing}} ({{MLSP}})},
  author = {Tolooshams, Bahareh and Dey, Sourav and Ba, Demba},
  year = {2018},
  archivePrefix = {arXiv},
  eprint = {1807.04734v1},
  eprinttype = {arxiv}
}

@inproceedings{Wohlberg2017,
  title={SPORCO: A Python package for standard and convolutional sparse representations},
  author={Wohlberg, Brendt},
  booktitle={Proceedings of the 15th Python in Science Conference, Austin, TX, USA},
  pages={1--8},
  year={2017}
}

```

---

> ### Author Response · Authors · 2020-11-25
> **Reviewer 1 Response**
>
> The authors would like to thank the reviewer for taking the time and careful reading of the manuscript and providing insightful comments that help improving the manuscript. Please see as follows how we have addressed the comments.
>
> ## Overall Assesment
> 1. The authors believe that this is not an equivalent framework. We have addressed this in the Appendix section.
> 2. The mentioned papers of [Isogawa2017] and [Ramzi2020] are related in their use of input-noise statistics in for varying deep-network threshold values at test-time. However, these methods do not perform blind denoising as they incorporate the true noise statistics. Our introduction of the adaptive thresholds is motivated by viewing our encoder as an unrolled sparse pursuit. We scale our thresholds with the estimated noise variance, as suggested by classical signal processing.
>  3. [Simon2019] proposed to use a large stride to reduce the mutual coherence between dictionary atoms. However, they use shift-averaging to compensate for the drop in the denoising performance incurred by using a large stride. We are able to reduce the mutual coherence with a small stride of 2 and frequency regularization, without shift-averaging. We choose not to test the performance with larger strides, because we want to avoid the shift-averaging operation, which would significantly increase the computational complexity.  Our method is also able to improve the learnt representation ([link](https://www.dropbox.com/s/swhxu9k22cm2hfh/acsccscnetfcdl.png?dl=0)).
>  4. The estimation formulation uses the diagonal-detail channel of a Wavelet transform which is not significantly affected by the coefficients from the signal as natural images do not have much information in the that channel. By utilizing the median as a robust estimator the effect of outliers (signal coefficients) are reduced. As discussed in [Mallat2008] (pg. 565), the estimator is unbiased.  Further discussion can be found in [Donoho1994, Donoho1995, Chang2000, Mallat2008].
>  5. Although our framework, CDLNet, is derived from an unrolled DL algorithm, it differs from other methods (such as [Wohlberg 2017] and [Moreau2019]) in that we learn the dictionary and approximate sparse-coding encoder in a supervised fashion. CDLNet differs from non deep-learning based DL methods in that we are able to offload much of the expensive iterative sparse-coding to training. Table 1 demonstrates the inefficiency of CSCNet's shift-averaging scheme and we show that better denoising performance can be achieved by FCDLNet with less computation. We believe that a computational complexity comparison between CDLNet and non deep-learning based dictionary learning algorithms is interesting but beyond the scope of this paper. The most expensive part of these algorithms is sparse coding. If the non-deep learning based models converge within the number of unrolled iterations of the deep-learning based ones, then the inference costs are comparable. Otherwise, the cost per iteration should be similar but the deep-learning models are expected to have faster inference time.
>  ## Some Questions
>  1. The CDLNet, ACSC network, and CSCNet claim greater interpretabilty over other deep-learning networks as they are derived from unrolled ISTA. This enables us to leverage classical results from signal processing in our network design. First, classical theory says the dictionary should have low mutual coherence to enable unique recovery. The proposed frequency regularization minimizes the mutual coherence, which leads to a better learnt dictionary (as in [link](https://www.dropbox.com/s/swhxu9k22cm2hfh/acsccscnetfcdl.png?dl=0)). Second, instead of using bias followed by ReLU as in a typical deep network, we use soft-thresholding, which is the proximal operator for L1-based regularization. Third, for denoising, the classical analysis reveals that the thresholds should be proportional to the noise variance and we leverage this to parameterize the threshold in our network so that a single trained network can work effectively over a large noise range. We have shown that shown that employing theoretical results from signal processing can lead to better blind-denoising performance.
>  2. The purpose of both constraints is to ensure a bounded norm of the dictionary atoms, which is satisfied in both cases. We've changed equation (7) to bound the norm of the effective dictionary for clarity.
>  3. Networks that employ the scaled thresholds do so with the estimated noise variance using the formula given in Section 2.3. The true noise value is never used in the network in neither training nor inference.
>  4. We train on the same dataset as CSCNet for roughly the same number of epochs. As we employ stride 2 without shift-processing, our network trains in roughly 1/4 of the time. With the GPU used in Table 1, our network trains in approximately 6 hours for K=10, M=64.

---

### Decision · Program_Chairs · 2021-01-07
**Final Decision**

**Decision:**

Reject

**Comment:**

The paper proposes a deep learning approach to blind image denoising based on deep unrolling. In particular, the proposed network is derived from convolutional sparse coding algorithms, which are unrolled, untied across layers and learned from data. The paper proposes a frequency domain regularization scheme in which the filters consist of a single analytically defined low-pass filter and a large collection of filters which are constrained to reside in the mid-to-high frequency ranges. It also proposes to tie the thresholds in the soft-thresholding stages of the learned network to estimates of the noise variance, making the proposed scheme more robust to variations in the noise level.

Pros and Cons:

[+] Having a single low-pass dictionary atom reduces redundancy (and potentially coherence) in the learned dictionary. This type of regularization may also reduce the time/data required to learn.

[+/-] Using noise estimators and a noise adaptive threshold renders the model more robust to variations in the noise level. This is important, since in most denoising applications the noise level is not known a-priori. As the reviewers note, the idea of tuning thresholds in an unrolled sparse coding method based on the noise level is not a novelty of the paper; the novelty here is coupling this with a wavelet-based estimate of the noise level.

[-] All three reviewers raise concerns regarding the novelty of the work compared to existing convolutional sparse coding-based neural networks. The structure of the network is similar; the main difference is the frequency restriction for learned atoms, which is enforced by prefiltering the learned atoms with a high-pass filter.

[-] The paper is not entirely clear in its motivation and argumentation. Reducing the coherence of the learned dictionary makes sense from the perspective of certain worst case results from sparse approximation. However, the coherence is a worst case quantity; moreover, certain approaches to coherence control (e.g., using large stride) control coherence at the expense of the expressiveness of the dictionary, and hence may not actually improve its ability to provide sparse reconstructions of natural signals. The proposed frequency domain regularization is a sensible approach to controlling coherence, since low-frequency atoms will tend to be highly coherent, but would benefit from a crisper analytical motivation.

[-] Reviewers found the experiments lacking in some regards. In particular, the paper only evaluates its proposals on synthetic experiments with Gaussian noise. While this is in line with some previous work on deep learning based denoising, more extensive and realistic experiments would have bolstered the paper's argument.

Overall, the paper makes a sensible proposal regarding the adaptivity to unknown noise levels, and introduces a potentially useful frequency-domain restriction on the learned filters in a CSC network. However, the reviewers did not find that the paper made a clear argument for the significance of these proposals, and raised other concerns regarding the clarity and experiments. The consensus of the reviewers is to recommend rejection.